# Clinical evaluation of the Diagnostic Analyzer for Selective Hybridization (DASH): A point-of-care PCR test for rapid detection of SARS-CoV-2 infection

Chad J. Achenbach[1,2,3]*, Matthew Caputo[1], Claudia Hawkins[1,2], Lauren C. Balmert[3], Chao Qi[4], Joseph Odorisio[1], Etienne Dembele[1], Alema Jackson[5], Hiba Abbas[5], Jennifer K. Frediani[6,7], Joshua M. Levy[6,8], Paulina A. Rebolledo[6,9], Russell R. Kempker[6,9], Annette M. Esper[6,10], Wilbur A. Lam[6,11], Greg S. Martin[6,10], Robert L. Murphy[1,2]

1 Havey Institute for Global Health, Feinberg School of Medicine, Northwestern University, Evanston, IL, United States of America, 2 Department of Medicine, Division of Infectious Diseases, Feinberg School of Medicine, Northwestern University, Evanston, IL, United States of America, 3 Department of Preventive Medicine, Feinberg School of Medicine, Northwestern University, Evanston, IL, United States of America, 4 Department of Pathology, Northwestern University, Evanston, IL, United States of America, 5 Access Community Health Network, Chicago, IL, United States of America, 6 Atlanta Center for Microsystems Engineered Point-of-Care Technologies, Atlanta, GA, United States of America, 7 Emory University Nell Hodgson Woodruff School of Nursing, Atlanta, GA, United States of America, 8 Emory University Department of Otolaryngology, Atlanta, GA, United States of America, 9 Emory University Division of Infectious Diseases, Atlanta, GA, United States of America, 10 Emory University Division of Pulmonary, Allergy, Critical Care and Sleep Medicine, Atlanta, GA, United States of America, 11 Emory University Department of Pediatrics, Atlanta, GA, United States of America

* c-achenbach@northwestern.edu

**Data Availability Statement:** All relevant data are within the paper and its Supporting Information files.

## Abstract

### Background

An ideal test for COVID-19 would combine the sensitivity of laboratory-based PCR with the speed and ease of use of point-of-care (POC) or home-based rapid antigen testing. We evaluated clinical performance of the Diagnostic Analyzer for Selective Hybridization (DASH) SARS-CoV-2 POC rapid PCR test.

### Methods

We conducted a cross-sectional study of adults with and without symptoms of COVID-19 at four clinical sites where we collected two bilateral anterior nasal swabs and information on COVID-19 symptoms, vaccination, and exposure. One swab was tested with the DASH SARS-CoV-2 POC PCR and the second in a central laboratory using Cepheid Xpert Xpress SARS-CoV-2 PCR. We assessed test concordance and calculated sensitivity, specificity, negative and positive predictive values using Xpert as the "gold standard".

### Results

We enrolled 315 and analyzed 313 participants with median age 42 years; 65% were female, 62% symptomatic, 75% had received ≥2 doses of mRNA COVID-19 vaccine, and

**Funding:** Funding provided by National Institutes for Health (NIH)/National Institute for Biomedical Imaging and Bioengineering (NIBIB)(https://www.nibib.nih.gov/), grant numbers U54EB027049-02S1 (PI is RLM) and U54EB027690 (Co-PIs are WAL and GSM). Minute Molecular Diagnostics, Inc. only provided DASH machines, cartridges, and nasal swabs. The funders had no role in study design, data collection and analysis, decision to publish, or preparation of the manuscript.

**Competing interests:** The authors have declared that no competing interests exist.

16% currently SARS-CoV-2 positive. There were concordant results for 307 tests indicating an overall agreement for DASH of 0.98 [95% CI 0.96, 0.99] compared to Xpert. DASH performed at 0.96 [95% CI 0.86, 1.00] sensitivity and 0.98 [95% CI 0.96, 1.00] specificity, with a positive predictive value of 0.85 [95% CI 0.73, 0.96] and negative predictive value of 0.996 [95% CI 0.99, 1.00]. The six discordant tests between DASH and Xpert all had high Ct values (>30) on the respective positive assay. DASH and Xpert Ct values were highly correlated (R = 0.89 [95% CI 0.81, 0.94]).

## Conclusions

DASH POC SARS-CoV-2 PCR was accurate, easy to use, and provided fast results (approximately 15 minutes) in real-life clinical settings with an overall performance similar to an EUA-approved laboratory-based PCR.

## Introduction

Rapid and accurate detection of SARS-CoV-2 infection has been a key component of the medical and public health response to the COVID-19 pandemic [1–3]. Despite advances in testing, we need better tools for early diagnosis and screening, particularly as we move toward "test and treat" strategies given several outpatient therapies for COVID-19 have been granted FDA emergency use authorization [4–7]. To date, the "gold standard" for accuracy, with highest analytic sensitivity, are nucleic acid amplification test (NAAT)-based technologies such as RT-PCR [8, 9]. However, PCR testing requires centralized laboratory equipment, a high level of technical expertise, is expensive, time consuming, and is unavailable in many rural or under-resourced settings. Current rapid point-of-care (POC) and home tests such as antigen or NAAT non-PCR-based technologies are simple to use, but with lower sensitivity resulting in a limited ability to detect individuals early in SARS-CoV-2 infection when pre-symptomatic or asymptomatic [10–23]. An ideal test for COVID-19 would have the accuracy of laboratory-based PCR testing combined with the speed and ease of use of POC rapid antigen testing [1].

To address this need, Minute Molecular Diagnostics, Inc. (m2dx.com), with support from the U.S. National Institutes of Health (NIH) National Institute of Biomedical Imaging and Bioengineering (NIBIB) Rapid Acceleration of Diagnostics (RADx[SM] Tech) program (https://www.nibib.nih.gov/covid-19/radx-tech-program), developed the Diagnostic Analyzer for Selective Hybridization (DASH), a sample-to-answer platform designed for POC COVID-19 testing in healthcare settings with potential applications in the community. DASH is a recently U.S. FDA EUA approved device (S1 File) that utilizes rapid PCR and microfluidic fabrication technology with uniquely designed cartridges and an analyzer that are easy to use, do not require technical expertise or specialized training, and produce results in approximately 15 minutes. DASH detects two targets in the N gene (N1 and N2) of SARS-CoV-2 in 40 PCR cycles. In laboratory testing, DASH was found to have analytic sensitivity at the same level as laboratory-based PCR with the ability to detect at least 150 copies/mL of SARS-CoV-2 RNA [24–26].

The primary objective of this clinical validation study was to test the performance and accuracy of the DASH POC test for diagnosis of SARS-CoV-2 infection among individuals with and without COVID-19 symptoms in outpatient settings.

## Methods

### Study design, population and recruitment

We performed a cross-sectional study from June to September 2021 of adult (18 years and older) participants with or without symptoms of COVID-19 who were tested for SARS-CoV-2 at one of four outpatient clinical sites. Asymptomatic participants self-reported no symptoms attributable to COVID-19 within 14 days prior to enrollment; however, prior to August 10, 2021, several participants were enrolled under previous FDA asymptomatic criteria of no symptoms within 48 hours. Symptomatic participants self-reported symptoms attributable to COVID-19 within the 24 hours prior to enrollment that started within the prior seven days. Symptoms attributable to COVID-19 include the following (as designated by CDC): fever or chills, cough, shortness of breath or difficulty breathing, fatigue, muscle or body aches, headache, new loss of taste or smell, sore throat, congestion or runny nose, nausea or vomiting, and diarrhea [27]. Participants were consented remotely via telephone utilizing Northwestern REDCap and emailed a survey link for data collection on symptoms, exposure details, demographics, vaccine history, risk factors for acquisition, and co-morbidities.

We had three phases of recruitment. Initially, from June to July 2021, we prospectively recruited and enrolled participants only at Chicago sites (Northwestern Memorial Hospital (NMH) Infectious Diseases Center (IDC) and NMH Clinical Research Unit (CRU)) regardless of risk or exposure and without recent known testing. Second, due to low COVID-19 prevalence in Chicago, in August 2021, we aimed to enhance test positivity by focusing recruitment on younger unvaccinated participants (18 to 45 years) and those with high-risk COVID-19 exposures, again only at Chicago sites (NMH IDC, NMH CRU, and Access Community Health Network). Finally, in September 2021, we performed enhanced recruitment where we included participants from a high COVID-19 prevalence outpatient sites in Georgia who were locally enrolled and consented in the Emory University "RADxtra COVID-19 Test Verification" study. Under our enhanced recruitment, we allowed for enrollment of individuals within seven days of testing positive for SARS-CoV-2 by a PCR-based assay. Clinical information from participants enrolled at the Emory RADxtra site was limited to data obtained through their study protocol.

### Sample collection and testing

After we obtained informed consent and participants completed REDCap-based on-line surveys, we performed a single research visit at each clinical site. At the visit, we obtained two bilateral anterior nasal swabs from each participant in a standardized process while wearing full PPE and abiding by all local institutional infection control policies regarding participant rooming, collection and handling of nasal swab samples from individuals known or suspected of having SARS-CoV-2 infection. For both anterior nasal swabs we used the Puritan Ultra 6" sterile elongated flock swab with plastic handle and dry transport tube (SKU #25-3606-U BT). To assure each swab was obtained from a "fresh" nostril, the study team clinician (physician or research nurse) first swabbed the left nostril anterior nares by inserting the swab at least 1 cm (0.5 inch) swirling the swab around the nasal wall for 10–15 seconds, then switched to the second swab and sampled the right nostril (same procedure as above), used the second swab to sample the left nostril, and finally switched back to the first swab to sample the right nostril. The first swab was placed into a designated plastic sleeve, inserted into a sealed biohazard bag, and transported for testing on DASH SARS-CoV-2 PCR (DASH) machine (Fig 1A) and the second swab placed into 3.0 mL of viral transport media (VTM; Remel MicroTest M4RT

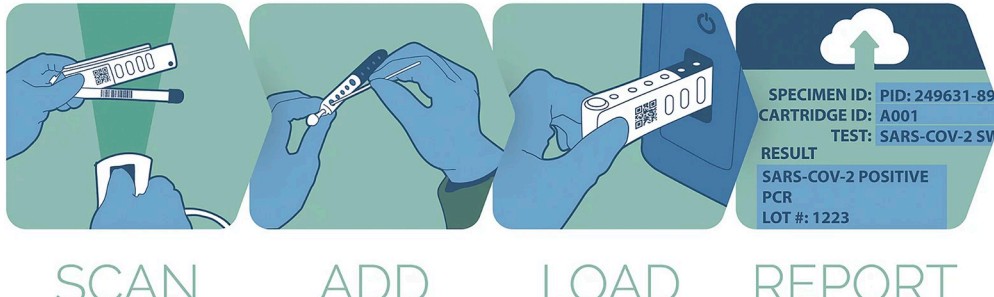

**Fig 1. A.** Diagnostic Analyzer for Selective Hybridization (DASH) PCR machine at point-of-care in clinical space. **B.** Diagnostic Analyzer for Selective Hybridization (DASH) PCR machine workflow: barcode on cartridge scanned, nasal swab added to cartridge and snapped at break point, cartridge loaded into DASH machine, and report generated on screen/printed after approximately 15 minutes.

Transport), lightly rotated for a few seconds, broken off, VTM tube sealed with cap, placed in a sealed biohazard bag, and transported for Cepheid Xpert Xpress SARS-Cov-2 RT-PCR (Xpert) testing. The DASH swab was inserted into the DASH cartridge and run on the machine as per the Quick Reference Guide (QRG) provided by Minute Molecular Diagnostics, Inc. at POC within 30 minutes of collection (Fig 1B). The nasal swab in VTM was transported in a sealed biohazard bag to the Northwestern Memorial Hospital (NMH) clinical microbiology laboratory and run on the Xpert testing platform within 4 hours of collection. Participants were given the results of only the Xpert standard PCR testing within 24 hours. If DASH and Xpert testing were discordant, then if residual VTM from Xpert testing was available, we performed tie breaker testing of residual VTM using Abbott Alinity M platform.

For research visits from May 22 to August 6, 2021, DASH testing machines were not available at the POC. During this period, the DASH nasal swab was placed into -80°C freezer in the NMH clinical microbiology laboratory within 4 hours of collection for temporary storage. Once DASH machines became available at POC, then frozen DASH samples were thawed to room temperature for at least an hour (no more than 4 hours) and placed into the DASH cartridge and assay run as per the QRG. Prior laboratory experiments found nasal swabs to be stable under these conditions with less than 6% difference in virus yield (S2 File).

The RADxtra study team at Emory also collected two bilateral anterior nasal swabs as per the same study procedure detailed above and then shipped de-identified DASH swabs and corresponding VTM overnight to Chicago on ice packs (2–8°C) in compliance with the Cepheid Xpert Xpress SARS-CoV-2 Emergency Use Authorized Instructions for Use [28]. Upon delivery to Northwestern, the direct nasal swab was run on the DASH machine as per QRG and the VTM nasal swab specimen was run on Xpert per the manufacturer instructions in the NM clinical microbiology laboratory. Again, prior laboratory experiments found nasal swabs to be stable under these conditions with less than 6% difference in virus yield (S1 File). We provided RADxtra investigators at Emory a unique identifier for each participant of samples/data sent to Northwestern. They kept this link to their study ID locally at Emory and none of the investigators or team at Northwestern had access to this linkage.

A usability survey was distributed to the DASH users after each use of the machine. This survey included questions on DASH complexity and ease of use with no specific training and only instruction from the QRG. For this initial study, we only analyzed the survey response from users who completed the survey after their first use of the DASH machine.

## Statistical analysis

Descriptive statistics were used to summarize demographics, exposure, and COVID-19 risk factors. Primary analyses calculated sensitivity and specificity from a 2X2 contingency table comparing DASH to the Xpert PCR. Specifically, sensitivity was calculated as the Pr(DASH+| Xpert+) or True Positives / (True Positives + False Negatives). Specificity was calculated as the Pr(DASH-|Xpert-) or True Negatives / (True Negatives + False Positives). In secondary analyses, we also calculated positive predictive value (PPV) and negative predictive value (NPV) using Bayes theorem assuming an *a priori* specified prevalence of 8%. Given the changing dynamic of the pandemic over the course of the study, we also estimated PPV and NPV for a range of prevalence estimates (1% to 30%). Confidence intervals for predictive values were

generated using a percentile bootstrap, with 1000 bootstrap replicates. Additionally, positive likelihood ratio (LR+) and negative likelihood ratio (LR-) were calculated as sensitivity/(1-specificity) and (1-sensitivity)/specificity, respectfully. 95% confidence intervals (CIs) were calculated for all estimates. Descriptive statistics were also used to summarize the DASH usability survey.

Within the subset of participants that tested positive on both Xpert and DASH, a Pearson correlation coefficient was used to assess the association between DASH and Xpert Ct values. A Bland-Altman plot visualized agreement between the two measures, using percentiles for the 95% CI to account for non-normality of differences.

All analyses were performed using R 4.1.1 (R Core Team, 2021) and the epiR (v2.0.36; Sergeant & Stevenson, 2021) package.

### Sample size calculations

Sample size considerations were based on estimates of sensitivity and specificity, with corresponding 95% CIs, necessary to meet minimum FDA performance criteria for Emergency Use Authorization (EUA). Specifically, a sample size of at least 275 participants was required to estimate sensitivity of 0.95 such that the lower bound of the 95% CI was greater than 0.76, and to estimate specificity of 0.98 such that the lower bound of the CI was greater than 0.95. Calculations assumed a population prevalence of positive SARS-CoV-2 of 8% and at least 30 participants overall in the study with a positive Xpert "gold standard" test result.

### Results

We enrolled 315 participants and obtained valid results from 313 participants at our four clinical sites with samples run by 15 different users on three DASH machines. Among our 313 participants analyzed, median age was 42 years, 65% were female, 75% had received two doses of mRNA COVID-19 vaccine (Pfizer or Moderna), 62% had COVID-19 symptoms, 14% had known exposure to someone with COVID-19, and 16% had known SARS-CoV-2 PCR positive testing prior to study enrollment (Table 1). Excluding participants from the Emory RADxtra site with limited clinical data (n = 116), 27% had co-morbidities.

We found a total of 51 participants tested positive on DASH and 49 participants tested positive on Xpert (Table 2). Summaries of DASH compared to Xpert testing are reported in Tables 2 and 3. Sensitivity of the DASH test was estimated at 0.96 [95% CI: 0.86, 1.00] and specificity was estimated at 0.98 [95% CI: 0.96, 1.00]. There was concordance of 307 tests indicating an overall agreement of 0.98 [95% CI: 0.96, 0.99]. The DASH test had a PPV of 0.85 [95% CI: 0.73, 0.96] and NPV of 0.996 [95% CI: 0.99, 1.00] as compared to the Xpert. Predictive value estimates for a range of infection prevalence rates are depicted in Fig 2 to assess performance across a wide range of COVID-19 prevalence. At the observed disease prevalence within the sample (16%), DASH demonstrated a PPV of 0.92 [95% CI 0.86, 0.98] and NPV of 0.99 [95% CI 0.98, 1.00], however this prevalence is likely an overestimate as known-positive participants were enrolled under our enhanced recruitment strategy. Within the subgroup of participants who tested positive on both Xpert and DASH (n = 45), Ct values for DASH and Xpert were highly correlated (Pearson correlation coefficient = 0.89 [95% CI 0.81, 0.94], Fig 3). Bland-Altman plot indicated a negative bias, such that Xpert Ct values were higher on average by 7.9 cycles (Fig 4).

Clinical and testing information for individual participants with discordant results (n = 6) are summarized in Table 4. We had two participants with negative DASH testing and positive Xpert testing and four participants with positive DASH testing and negative Xpert testing. All six of these individuals had at least one COVID-19 symptom and/or known positive PCR

**Table 1. Characteristics of 313 participants with a valid Diagnostic Analyzer for Selective Hybridization (DASH) SARS-CoV-2 PCR test result.**

| Characteristic | N (%) or Median (IQR) |
|---|---|
| **Clinical Site** | |
| Total | 313 (100) |
| NMH IDC | 112 (36) |
| NMH CRU | 71 (23) |
| Access Community Health Network | 14 (5) |
| Emory RADxtra Clinics | 116 (37) |
| **Age in years** | 42 (32, 57) |
| **Sex Assigned at Birth** | |
| Female | 204 (65) |
| Male | 107 (34) |
| Not Reported | 2 (1) |
| **Race**\* | |
| White | 155 (79) |
| Asian | 17 (9) |
| Black or African American | 17 (9) |
| Other or unknown | 8 (4) |
| **Ethnicity**\* | |
| Non-Hispanic or unknown | 187 (95) |
| Hispanic | 10 (5) |
| **COVID-19 Vaccine Doses** | |
| 0 | 55 (18) |
| 1 | 23 (7) |
| 2 | 229 (73) |
| Vaccinated unknown doses | 6 (2) |
| **Vaccine Manufacturer**\*\*\*\* | |
| Pfizer | 145 (56) |
| Moderna | 89 (34) |
| Johnson & Johnson | 15 (6) |
| Unknown type | 9 (4) |
| **Days since last vaccine dose**\*\*\*\* | 140 (101, 171) |
| **Samples from Enrichment** | 51 (16) |
| **Known SARS-CoV-2 Exposure within 2 weeks** | 28 (9) |
| **Known SARS-CoV-2 Positive (PCR positive within 7 days)** | 50 (16) |
| **COVID-19 Symptomatic** | |
| Yes | 195 (62) |
| No | 118 (38) |
| **Individual COVID-19 Symptoms**\*\* | |
| Congestion | 78 (40) |
| Cough | 78 (40) |
| Headache | 68 (35) |
| Fatigue | 60 (31) |
| Fever/Chills | 59 (30) |
| Sore/Scratchy Throat | 53 (27) |
| Vomiting/Nausea/Diarrhea | 51 (26) |
| Myalgias | 49 (25) |
| Loss of Taste/Smell | 37 (19) |

*(Continued)*

**Table 1.** (Continued)

| Characteristic | N (%) or Median (IQR) |
|---|---|
| **Clinical Site** | |
| Shortness of Breath | 25 (13) |
| Arthralgias | 25 (13) |
| Abdominal Pain | 20 (10) |
| Photophobia | 15 (8) |
| **Medical History**\* | |
| Any co-morbidity | 52 (26) |
| Hypertension | 22 (11) |
| Asthma | 9 (5) |
| Diabetes | 7 (4) |
| Cancer | 6 (3) |
| Immunodeficiency (Not HIV) | 5 (3) |
| Coronary Artery Disease | 4 (2) |
| Anemia | 4 (2) |
| Other\*\*\* | 9 (5) |

\*Excluding 116 participants from Emory RADxtra site as they did not provide this demographic or clinical information (n = 197).

\*\*Among those reporting symptoms (n = 195); categories are not mutually exclusive.

\*\*\*Others were COPD (n = 3), renal failure (n = 2), stroke/TIA (n = 2), HIV (n = 1), and MI (n = 1).

\*\*\*\*Among those that received at least 1 vaccine dose prior to testing (n = 258).

Abbreviations: NMH IDC, Northwestern Memorial Hospital Infectious Diseases Center; NMH CRU, Northwestern Memorial Hospital Clinical Research Unit.

testing for COVID prior to enrollment. Both false negatives had high Ct values (>35 cycles) on Xpert testing, and 3 of the 4 false positives had high Ct values (>35 cycles) or an indeterminate result (one of two SARS-CoV-2 gene targets detected) on the tiebreaker assay. Five of these individuals with discordant results were late in their course of infection (at least 4 days from onset of symptoms or prior positive PCR testing).

Ten of fifteen users completed our usability survey after their first time running a participant sample on the DASH machine. When asked to rate their agreement with the statement "I found this DASH test was easy to use", 40% strongly agreed (4 users), 50% agreed (5 users), and one user was neutral. When asked, "would you recommend DASH test to someone else?", 50% responded "definitely yes" (5 users) 40% responded "probably yes" (4 users), and one user did not respond.

## Discussion

In this cross-sectional clinical study, we found that the DASH SARS-CoV-2 PCR was easy to use, fast (time from placement of direct nasal swab into cartridge until result of approximately 15 minutes) and performed with excellent test characteristics (sensitivity of 0.96, specificity of 0.98, and overall agreement of 0.98) compared to a "gold standard" laboratory-based PCR in real-life outpatient healthcare settings. From 313 participants with valid results, there were only six discordant results and each of these participants had at least one clinical symptom of COVID-19 or known positive PCR tests. DASH was negative and missed only two infected individuals; however, they both had high Ct values (>39) on standard PCR and thus were unlikely to be contagious [29–32]. In contrast, DASH detected four individuals with

**Table 2. Summary of Diagnostic Analyzer for Selective Hybridization (DASH) PCR results compared to Cepheid Xpert Xpress (Xpert) PCR results.**

|                | Xpert Positive | Xpert Negative | Total |
|----------------|:--------------:|:--------------:|:-----:|
| DASH Positive  | 47             | 4*             | 51    |
| DASH Negative  | 2*             | 260            | 262   |
| Total          | 49             | 264            | 313   |

*See Table 4 for summary of clinical and testing details from 6 participants with discordant DASH and Xpert results.

SARS-CoV-2 that were missed by Xpert, with only one of these as a potential false positive as RNA was not detected on tie breaker PCR testing and the participant only had dyspnea without other symptoms typical of COVID-19. Discordance only occurred at high Ct values and is consistent with known limitations of SARS-CoV-2 PCR testing assays early or late in infection when virus levels are low [8, 9, 33, 34]. Thus, we believe that DASH performed equivalent to laboratory-based PCR testing and displays its potential as a rapid and highly sensitive POC option to supplement or replace current PCR platforms in real-life clinical settings and eventually the community.

DASH improves upon current POC non-PCR NAAT technologies for SARS-CoV-2 such as Abbott ID NOW. In several studies, the sensitivity of the ID NOW test was found to be between 0.75–0.98 depending on number of low positive samples (Ct value >35 on standard PCR assays) [35–40]. Limiting to positive Xpert with Ct values from 35 to 39, DASH detected all low positive samples, thus displaying an ability to potentially detect virus earlier in the course of SARS-CoV-2 infection compared to ID NOW. This has become an important public health issue when testing highly vaccinated populations and those with emerging variants, such as Omicron, who may have a shorter window of SARS-CoV-2 detection or lower levels of virus after high-risk exposure [41–44].

In addition, DASH is easy to use (Clinical Laboratory Improvement Amendments (CLIA)-waived) and does not require technical or laboratory training to run the machine. Our research team of 15 different users had neither formal training nor laboratory expertise and were able to perform DASH testing with valid results from 99.4% of participants sampled. One downside to DASH, and several other POC SARS-CoV-2 diagnostics, is inability to perform repeat testing on the same sample. In the event of an invalid result, patients will be required to provide a second specimen for DASH testing. Only using instruction from the QRG, 90% of our users assessed DASH to be easy to use and they all recommended DASH to others who might be doing COVID testing. We plan to expand these assessments as more users have access to DASH and perform qualitative research to optimize DASH usability in clinical and non-clinical settings.

Correlation analyses (see Fig 3) between DASH and Xpert Ct values demonstrated a strong association; however, Bland-Altman plots (Fig 4) indicated DASH PCR Ct values were systematically lower than Xpert PCR. This occurred due to several technical differences between DASH and Xpert in sample processing standards leading to higher virus concentrations with DASH. DASH and Xpert PCRs both detect two SARS-CoV-2 targets; however, DASH does this in the same fluorescent channel whereby Xpert splits molecular material between two separate channels. Also, for DASH, the nasal swab was directly inserted into the cartridge whereas Xpert swab was placed in 3 mL of VTM, and then 300 μl of liquid media added to the cartridge. Finally, DASH processes approximately 50% of the specimen and Xpert only processes 10%. The goal of comparing Ct values between these technologies was to assess whether DASH can provide meaningful semi-quantitative viral load measurements, similar to laboratory-based

**Table 3. Diagnostic Analyzer for Selective Hybridization (DASH) POC PCR diagnostic performance and accuracy compared to Cepheid Xpert Xpress PCR as "gold standard".**

| Statistic | n/N | Estimate [95% CI] |
|---|---|---|
| Apparent Prevalence | 49/313 | 0.16 [0.12, 0.21] |
| Sensitivity | 47/49 | 0.96 [0.86, 1.00] |
| Specificity | 260/264 | 0.98 [0.96, 1.00] |
| Overall Diagnostic Accuracy | 307/313 | 0.98 [0.96, 0.99] |
| Positive Predictive Value* | | 0.85 [0.73, 0.96] |
| Negative Predictive Value* | | 0.996 [0.99, 1.00] |
| Positive Likelihood Ratio | | 63.31 [23.90, 167.71] |
| Negative Likelihood Ratio | | 0.04 [0.01, .16] |

*Estimated using Bayes' theorem assuming a prevalence of 8%.

PCR, but at POC and quick enough to potentially influence clinical or public health guidance. A more accurate comparison between DASH and Xpert would involve preparing standard specimens with known virus concentrations and processing them with both technologiesc [45]. Unfortunately, this was outside of the scope of this clinical validation study evaluating SARS-CoV-2 detection using DASH.

Our study had several limitations. First, the cross-sectional design did not allow us to understand test characteristics of DASH over time and throughout the course of SARS-CoV-2 infection within individuals. Second, some of the specimens were tested on DASH after short-term (less than 7 days) in frozen storage. However, given the stability observed in frozen samples from viability testing (see S2 File), we do not believe this impacted our findings. Third, we were unable to assess the public health safety impact of DASH testing since the test was experimental and we were unable to give participants their result. Now that DASH has been FDA EUA approved, we have planned follow-up clinical validation studies where this can be addressed. Fourth, most of

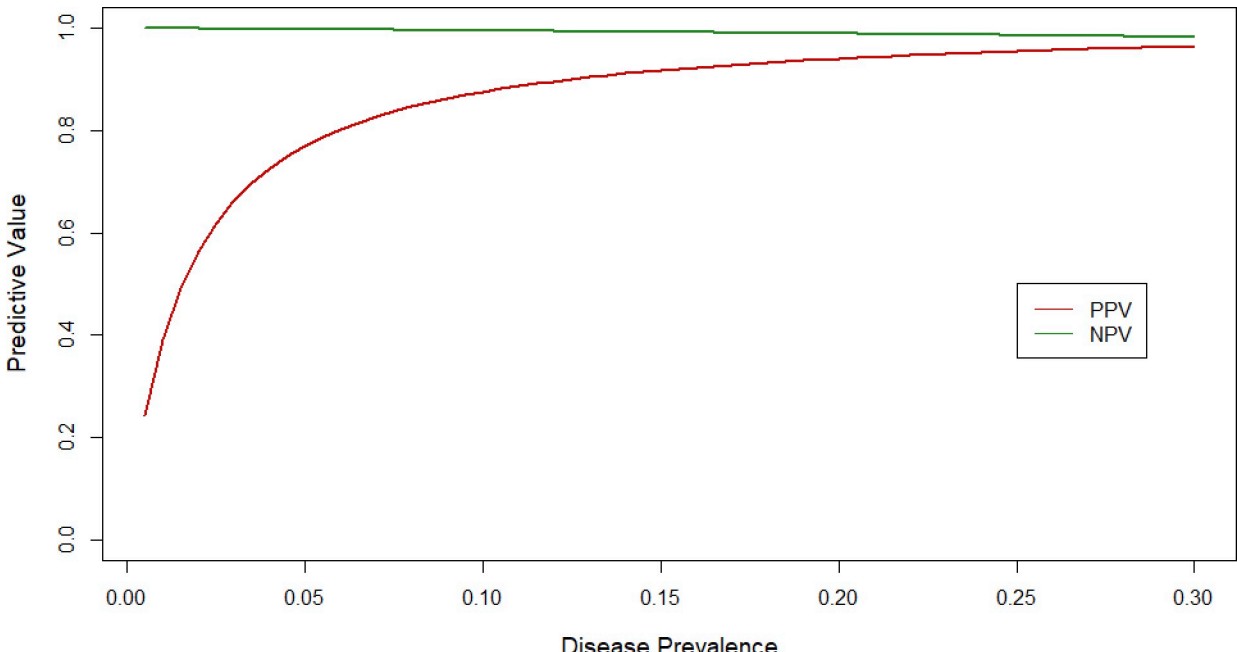

**Fig 2. Positive (red line) and negative (green line) predictive values of Diagnostic Analyzer for Selective Hybridization (DASH) SARS-CoV-2 PCR test by varying COVID-19 prevalence.**

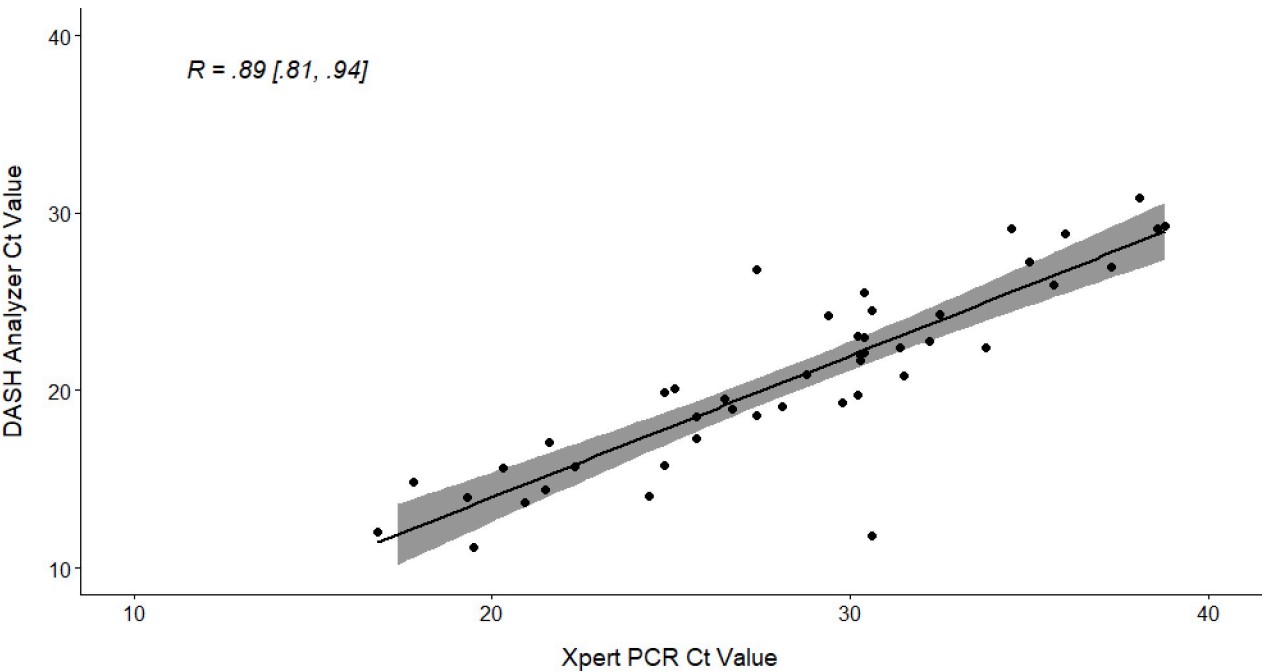

**Fig 3. Correlation analyses for comparison of Diagnostic Analyzer for Selective Hybridization (DASH) PCR and Cepheid Xpert Xpress PCR cycle threshold (Ct) values.**

our participants were symptomatic and many known to be SARS-CoV-2 positive with only one positive sample from an asymptomatic participant. Additionally, our participants were young, highly vaccinated, and without major co-morbidities including immune suppression. Thus, DASH test characteristics are less certain for those at greatest risk for severe COVID-19 or among

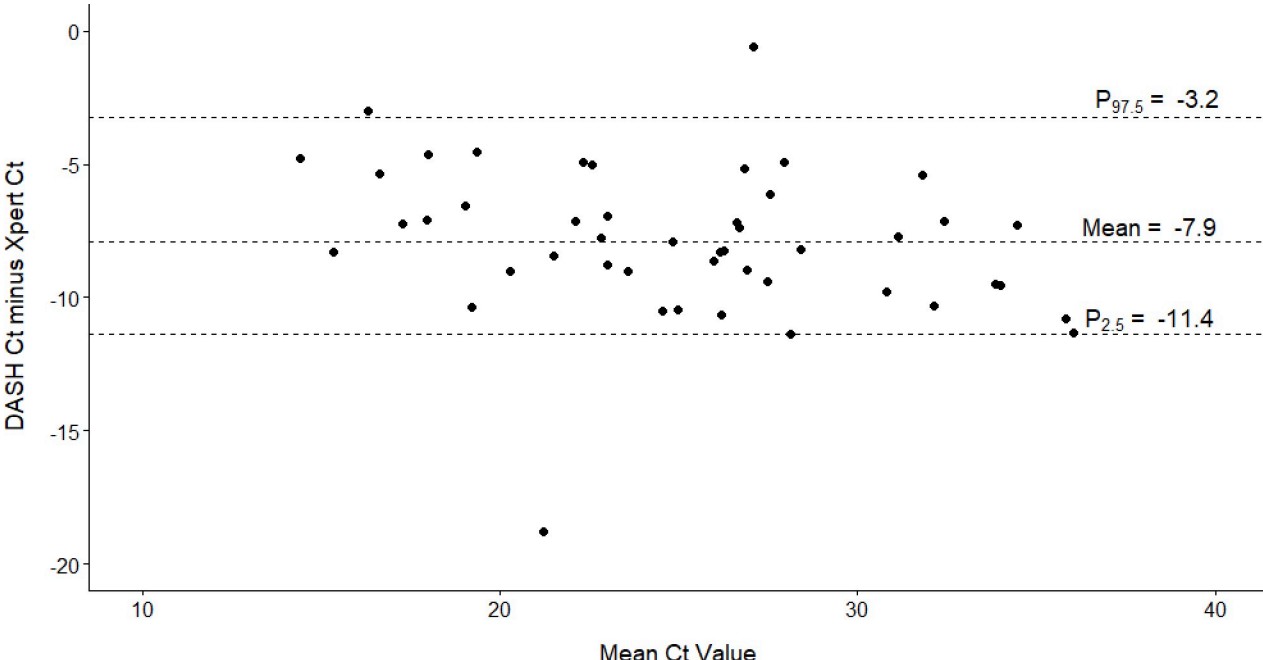

**Fig 4. Bland-Altman analyses for comparison of Diagnostic Analyzer for Selective Hybridization (DASH) PCR and Cepheid Xpert Xpress PCR cycle threshold (Ct) values.**

**Table 4. Summary of discordant results between Diagnostic Analyzer for Selective Hybridization (DASH) and Cepheid Xpert Xpress (Xpert) PCR testing.**

| Site | Xpert PCR (Ct if positive) | DASH PCR (Ct if positive) | Tiebreaker Result (Ct if positive) * | Known Positive Platform (days prior, Ct) | Symptoms; Onset; Vaccine doses (type) |
|---|---|---|---|---|---|
| Emory | Detected (42.5) | Not Detected | Not Done; sample discarded | Roche Cobas 6800 (unknown) | cough, HA, sore throat, fatigue, diarrhea, shortness of breath; 4 days; unvaccinated |
| NMH IDC | Detected (39.8) | Not Detected | Not Done; sample discarded | Abbott Alinity M (5, 34.1) | no symptoms; NA; 2 doses (Pfizer) |
| NMH IDC | Not Detected | Detected (34.1) | Indeterminate | Cepheid Xpert Xpress (2, 38.2) | cough, sinus congestion, loss of taste and smell; 17 days; 1 dose (J&J) |
| NMH IDC | Not Detected | Detected (29.9) | Detected (36.2) | Thermo Fisher TaqPath COVID-19 Combo Kit (8, Ct unknown) | scratchy throat, Sore throat, cough, runny nose, Fever/chills, loss of taste or smell; 13 days; 1 dose (J&J) |
| Emory | Not Detected | Detected (33.2) | Detected (36.8) | Not done | HA, sore throat, abdominal pain, loss of sense of taste or smell; 4 days; 2 doses (Pfizer) |
| Emory | Not Detected | Detected (33.4) | Not Detected | Abbott Alinity M (unknown) | shortness of breath; 1 day; 2 doses (Pfizer) |

*All tiebreaker assays run on Abbott Alinity M platform.

Abbreviations: NMH ID, Northwestern Memorial Hospital Infectious Diseases Center; HA, headache; J&J, Johnson & Johnson.

certain special populations. Although, other PCR-based SARS-CoV-2 testing technologies have generally performed well across many different high-risk patient groups [46–51]. Finally, we obtained anterior nasal swabs for this study and did not do a comparison of nasopharyngeal swabs. Nasopharyngeal swabs require healthcare workers for collection and are considered the most sensitive specimen collection method for SARS-CoV-2 [52–54]. Given a preference for less invasive nasal testing, DASH portability and potential for future use in community non-health-care settings, we decided to perform anterior nasal sampling for this study and the DASH FDA EUA application. Additional sample types including saliva, nasopharyngeal and oropharyngeal swabs will be considered and evaluated in further clinical studies.

DASH is an important advancement in molecular POC diagnostics for SARS-CoV-2 with potential applications beyond many non-PCR NAAT technologies. DASH could decentralize PCR diagnostics by bringing molecular testing to a wider range of community settings in the U.S. and throughout the world. We need highly sensitive and easy to use technologies with the ability to quantify virus for monitoring SARS-CoV-2 infection dynamics and when to safely end isolation. Further research will allow us to understand how DASH could be utilized for detecting emerging SARS-CoV-2 variants, community surveillance, or large non-health care screening of travelers, students, educators, and front-line workers.

## Supporting information

**S1 Data.**
(CSV)

**S1 File.**
(PDF)

**S2 File.**
(PDF)

## Acknowledgments

These findings are presented on behalf of the study participants who generously gave their time, samples, and information for this research. We would also like to thankfully

acknowledge all staff and faculty in the NMH IDC, NMH CRU, Access Community Health Network, Emory RADxtra clinics, and NMH microbiology laboratory who contributed to this study. Emory investigators would like to acknowledge the following individual team members for their assistance with this study: Tamara Wesley, Tim Thurman, Kristi Godbolt, Anna Wood, Adrianna Westbrook, Leona Wells, Julie Sullivan, Cheryl L. Stone, and Jared O'Neal. Northwestern and Access investigators would like to acknowledge the following individual team members for their assistance with this study: Kate Klein, Camille Bundy, Mackenzie Furnari-Stickney, Amelia Kelly, Jehannaz Dastoor, Michael Govern, Carol Govern, Kristen Weber, Elizabeth Christian, Adrianna Quintana, Patricia Helbin, and Vinay Durbhakula.

## Author Contributions

**Conceptualization:** Chad J. Achenbach, Claudia Hawkins, Lauren C. Balmert, Chao Qi, Robert L. Murphy.

**Data curation:** Matthew Caputo, Lauren C. Balmert.

**Formal analysis:** Matthew Caputo, Lauren C. Balmert.

**Funding acquisition:** Wilbur A. Lam, Greg S. Martin, Robert L. Murphy.

**Investigation:** Chad J. Achenbach, Matthew Caputo, Claudia Hawkins, Chao Qi, Joseph Odorisio, Etienne Dembele, Alema Jackson, Hiba Abbas, Jennifer K. Frediani, Joshua M. Levy, Paulina A. Rebolledo, Russell R. Kempker, Annette M. Esper, Wilbur A. Lam, Greg S. Martin.

**Methodology:** Chad J. Achenbach, Lauren C. Balmert, Chao Qi.

**Project administration:** Chad J. Achenbach, Claudia Hawkins, Joseph Odorisio, Etienne Dembele, Alema Jackson, Hiba Abbas, Jennifer K. Frediani, Joshua M. Levy, Paulina A. Rebolledo, Russell R. Kempker, Annette M. Esper, Wilbur A. Lam, Greg S. Martin.

**Supervision:** Chad J. Achenbach, Claudia Hawkins, Lauren C. Balmert, Chao Qi, Wilbur A. Lam, Greg S. Martin, Robert L. Murphy.

**Visualization:** Matthew Caputo.

**Writing – original draft:** Chad J. Achenbach, Matthew Caputo.

**Writing – review & editing:** Claudia Hawkins, Lauren C. Balmert, Chao Qi, Joseph Odorisio, Etienne Dembele, Alema Jackson, Hiba Abbas, Jennifer K. Frediani, Joshua M. Levy, Paulina A. Rebolledo, Russell R. Kempker, Annette M. Esper, Wilbur A. Lam, Greg S. Martin, Robert L. Murphy.

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
