## [Decision Letter · Decision Letter 0]

8 Mar 2022

PONE-D-22-03445Clinical evaluation of the Diagnostic Analyzer for Selective Hybridization (DASH): a point-of-care PCR test for rapid detection of SARS-CoV-2 infectionPLOS ONE

Dear Dr. Achenbach,

Thank you for submitting your manuscript to PLOS ONE. After careful consideration, we feel that it has merit but does not fully meet PLOS ONE’s publication criteria as it currently stands. Therefore, we invite you to submit a revised version of the manuscript that addresses the points raised during the review process. I have received reviews of your manuscript. While your paper addresses an interesting question, the reviewers stated several concerns that need to be addressed carefully.  Please see reviewers' insightful information.  One of the main concerns which I shared is the lack of information about DASH.  Any background, principle, reference(s) that could help readers understand the mystery of the blue box? Specific comments:1. Suggest combine Fig. 1 & 2 into one 2. Results, first paragraph, line 5 (page 8):  Suggest change "... exposure to someone COVID-19" to "...exposure to someone with COVID-19"3. Discussion, page 15:   Plesae rephrase the last sentence for clarity. 4. Discussion, page 16, second paragraph:  Regarding the dynamic detection range, has this being done by the manufacture or during CLIA certification?

We look forward to receiving your revised manuscript.

Kind regards,

Baochuan Lin, Ph.D.

Academic Editor

PLOS ONE

Journal Requirements:

Reviewers' comments:

Reviewer's Responses to Questions

**Comments to the Author**

1. Is the manuscript technically sound, and do the data support the conclusions?

Reviewer #1: Partly

Reviewer #2: Yes

2. Has the statistical analysis been performed appropriately and rigorously? 

Reviewer #1: Yes

Reviewer #2: Yes

3. Have the authors made all data underlying the findings in their manuscript fully available?

Reviewer #1: No

Reviewer #2: Yes

4. Is the manuscript presented in an intelligible fashion and written in standard English?

Reviewer #1: Yes

Reviewer #2: Yes

5. Review Comments to the Author

Reviewer #1: The authors present a methods comparison study using samples from 313 patients for SARS-CoV-2 testing using either a new device, "Diagnostic Analyzer for Selective Hybridization (DASH): a point-of-care PCR test for rapid detection of SARS-CoV-2 infection" or as a standard the "Cepheid Xpert Xpress SARS-CoV-2 PCR". 16% of participants were SARS-CoV-2 positive in the reference method. The DASH measurements were partly done from frozen samples.

The authors found good agreement of the test results from the experimental device with the reference method.

POC SARS-CoV-2 testing is necessary in the clinical context to avoid spreading of COVID-19 and therefore new developments are valuable and need to be tested in prospective studies. I have some comments and questions:

1. The DASH device needs to be described in more detail

2. The comparison of raw Ct-values is interesting (fig. 4/5), but the authors need to measure virus preparation standards with known virus concentrations to compare semi quantitative virus concentrations rather than the Ct values which depend on the specific instruments. It is nowadays necessary to know, whether the virus load is below 10exp6. See "SARS-CoV-2 screening in patients in need of urgent inpatient treatment in the Emergency Department (ED) by digitally integrated point-of-care PCR: a clinical cohort study. by Möckel M, Bolanaki M, Hofmann J, Stein A, Hitzek J, Holert F, Fischer-Rosinský A, Slagman A.

Diagn Microbiol Infect Dis. 2022 Jan 16;102(4):115637. doi: 10.1016/j.diagmicrobio.2022.115637." for details.

3. Part of the comparison needs to be the sample management and handling at the point of care. The authors should give more details here. Are there regulatory issues and safety requirements to process potentially infectious samples?

4. The data availability statement is incomplete. There are no raw data available and also no source for it.

5. The authors provide data on frozen sample stability. They should in addition discuss as a limitation that the measurement from frozen samples is not comparable with acute sample measurement.

6. The authors used anterior nasal swabs which are less sensitive compared to deep nasophayryngeal swabs. This should be discussed.

7. The participants did not get results from the experimental device. Therefore, no safety consideration could be assessed. I believe that the results are therefore preliminary and an additional study with the experimental device for clinical use and backup comparison with a reference method is necessary.

8. I do not understand the figure 3. Is this a simulation? The authors did not gather data from different populations. I think this figure is superfluous.

9 The authors should consequently use "SARS-CoV-2" if they address the virus and "COVID-19" only if they address the disease. Nobody is "COVID-19 positive" (see abstract for example)

Reviewer #2: Dr. Chad Achenbach performed a prospective evaluation of a POC-type molecular platform (DASH) and its SARS-CoV-2 assay with anterior nasal samples. While there are some problems associated with conducted a DASH evaluation due to the low prevalence of COVID-19 during the study period in the target cites, the current protocol is well designed.

Introduction

1. A reference is needed for the following text: “DASH utilizes rapid PCR and microfluidic fabrication technology with uniquely designed cartridges and an analyzer that are easy to use, do not require technical expertise or specialized training, and produce results in approximately 15 minutes. In laboratory testing, DASH was found to have analytic sensitivity at the same level as laboratory-based PCR with the ability to detect at least 100 copies/mL of SARS-CoV-2 RNA”.

2. Please add more information on the DASH SARS-CoV-2 assay, such as the target genes, PCR cycles, etc.

Methods

1. Page 5, viral transport media (VTM): Please add the volume of VTM and brand name (including company information).

2. What kind of swabs were used in the current study?

Methods/Results/Discussion

1. How about invalid results in this study? A DASH re-test cannot be performed with the initially obtained samples. What do technicians do if the test results are not obtained properly due to invalidity? When emphasizing the merits of DASH in the Discussion, you should also describe the demerits. For

2. The current evaluation was only performed with anterior nasal samples. If nasopharyngeal samples or saliva are used for the evaluation, false-positive results might be obtained. You should mention that current investigation was only performed with anterior nasal samples.

3. In this study, most of the positive samples were obtained from patients with preceding positive results for SARS-CoV-2 according to other molecular assays. The current results should be validated in a further study. You should mention the limitations in the Discussion.

6. PLOS authors have the option to publish the peer review history of their article (what does this mean?). If published, this will include your full peer review and any attached files.

Reviewer #1: No

Reviewer #2: No

---

## [Author Response · Author response to Decision Letter 0]

22 Apr 2022

Responses and rebuttals to each specific reviewer and editor comment are found in the 'Response to Reviewers' document uploaded as part of this revision submission process.

---

## [Decision Letter · Decision Letter 1]

3 Jun 2022

Clinical evaluation of the Diagnostic Analyzer for Selective Hybridization (DASH): a point-of-care PCR test for rapid detection of SARS-CoV-2 infection

PONE-D-22-03445R1

Dear Dr. Achenbach,

We’re pleased to inform you that your manuscript has been judged scientifically suitable for publication and will be formally accepted for publication once you corrected the following: 1) line 20, delete "combined"; 2) line 28 change “gold standard.” to “gold standard”.; 3) line 51, change "are" to "such as"; 4) line 144, please spell out IFU; and it meets all outstanding technical requirements.

Kind regards,

Baochuan Lin, Ph.D.

Academic Editor

PLOS ONE

Additional Editor Comments (optional):

Reviewers' comments:

Reviewer's Responses to Questions

**Comments to the Author**

1. If the authors have adequately addressed your comments raised in a previous round of review and you feel that this manuscript is now acceptable for publication, you may indicate that here to bypass the “Comments to the Author” section, enter your conflict of interest statement in the “Confidential to Editor” section, and submit your "Accept" recommendation.

Reviewer #2: All comments have been addressed

2. Is the manuscript technically sound, and do the data support the conclusions?

Reviewer #2: Yes

3. Has the statistical analysis been performed appropriately and rigorously? 

Reviewer #2: Yes

4. Have the authors made all data underlying the findings in their manuscript fully available?

Reviewer #2: Yes

5. Is the manuscript presented in an intelligible fashion and written in standard English?

Reviewer #2: Yes

6. Review Comments to the Author

Reviewer #2: (No Response)

7. PLOS authors have the option to publish the peer review history of their article (what does this mean?). If published, this will include your full peer review and any attached files.

Reviewer #2: No

---

## [Editor Report · Acceptance letter]

8 Jun 2022

PONE-D-22-03445R1 

Clinical evaluation of the Diagnostic Analyzer for Selective Hybridization (DASH): a point-of-care PCR test for rapid detection of SARS-CoV-2 infection 

Dear Dr. Achenbach:

I'm pleased to inform you that your manuscript has been deemed suitable for publication in PLOS ONE. Congratulations! Your manuscript is now with our production department. 

Kind regards, 

on behalf of

Dr. Baochuan Lin 

Academic Editor

PLOS ONE